# Online Learning of Quantum States

**Scott Aaronson**
UT Austin *
aaronson@cs.utexas.edu

**Xinyi Chen**
Google AI Princeton †
xinyic@google.com

**Elad Hazan**
Princeton University and Google AI Princeton
ehazan@cs.princeton.edu

**Satyen Kale**
Google AI, New York
satyenkale@google.com

**Ashwin Nayak**
University of Waterloo ‡
ashwin.nayak@uwaterloo.ca

## Abstract

Suppose we have many copies of an unknown $n$-qubit state $\rho$. We measure some copies of $\rho$ using a known two-outcome measurement $E_1$, then other copies using a measurement $E_2$, and so on. At each stage $t$, we generate a current hypothesis $\omega_t$ about the state $\rho$, using the outcomes of the previous measurements. We show that it is possible to do this in a way that guarantees that $|\mathrm{Tr}(E_i\omega_t) - \mathrm{Tr}(E_i\rho)|$, the error in our prediction for the next measurement, is at least $\varepsilon$ at most $\mathrm{O}(n/\varepsilon^2)$ times. Even in the "non-realizable" setting—where there could be arbitrary noise in the measurement outcomes—we show how to output hypothesis states that incur at most $\mathrm{O}(\sqrt{Tn}\,)$ excess loss over the best possible state on the first $T$ measurements. These results generalize a 2007 theorem by Aaronson on the PAC-learnability of quantum states, to the online and regret-minimization settings. We give three different ways to prove our results—using convex optimization, quantum postselection, and sequential fat-shattering dimension—which have different advantages in terms of parameters and portability.

## 1 Introduction

*State tomography* is a fundamental task in quantum computing of great practical and theoretical importance. In a typical scenario, we have access to an apparatus that is capable of producing many copies of a quantum state, and we wish to obtain a description of the state via suitable measurements. Such a description would allow us, for example, to check the accuracy with which the apparatus constructs a specific target state.

How many single-copy measurements are needed to "learn" an unknown $n$-qubit quantum state $\rho$? Suppose we wish to reconstruct the full $2^n \times 2^n$ density matrix, even approximately, to within $\varepsilon$ in trace distance. If we make no assumptions about $\rho$, then it is straightforward to show that the number of measurements needed grows exponentially with $n$. In fact, even when we allow joint measurement of multiple copies of the state, an exponential number of copies of $\rho$ are required (see,

e.g., O'Donnell and Wright [2016], Haah et al. [2017]). (A "joint measurement" of two or more states on disjoint sequences of qubits is a *single* measurement of all the qubits together.)

Suppose, on the other hand, that there is some probability distribution $\mathcal{D}$ over possible yes/no measurements, where we identify the measurements with $2^n \times 2^n$ Hermitian matrices $E$ with eigenvalues in $[0,1]$. Further suppose we are only concerned about learning the state $\rho$ well enough to predict the outcomes of *most* measurements $E$ drawn from $\mathcal{D}$—where "predict" means approximately calculating the probability, $\mathrm{Tr}(E\rho)$, of a "yes" result. Then for how many (known) sample measurements $E_i$, drawn independently from $\mathcal{D}$, do we need to know the approximate value of $\mathrm{Tr}(E_i\rho)$, before we have enough data to achieve this?

Aaronson [2007] proved that the number of sample measurements needed, $m$, grows only *linearly* with the number of qubits $n$. What makes this surprising is that it represents an exponential reduction compared to full quantum state tomography. Furthermore, the prediction strategy is extremely simple. Informally, we merely need to find any "hypothesis state" $\omega$ that satisfies $\mathrm{Tr}(E_i\omega) \approx \mathrm{Tr}(E_i\rho)$ for all the sample measurements $E_1, \ldots, E_m$. Then with high probability over the choice of sample measurements, that hypothesis $\omega$ necessarily "generalizes", in the sense that $\mathrm{Tr}(E\omega) \approx \mathrm{Tr}(E\rho)$ for most additional $E$'s drawn from $\mathcal{D}$. The learning theorem led to followup work including a full characterization of quantum advice (Aaronson and Drucker [2014]); efficient learning for stabilizer states (Rocchetto [2017]); the "shadow tomography" protocol (Aaronson [2018]); and recently, the first experimental demonstration of quantum state PAC-learning (Rocchetto et al. [2017]).

A major drawback of the learning theorem due to Aaronson is the assumption that the sample measurements are drawn *independently* from $\mathcal{D}$—and moreover, that the same distribution $\mathcal{D}$ governs both the training samples, and the measurements on which the learner's performance is later tested. It has long been understood, in computational learning theory, that these assumptions are often unrealistic: they fail to account for adversarial environments, or environments that change over time. This is precisely the state of affairs in current experimental implementations of quantum information processing. Not all measurements of quantum states may be available or feasible in a specific implementation, *which* measurements are feasible is dictated by Nature, and as we develop more control over the experimental set-up, more sophisticated measurements become available. The task of learning a state prepared in the laboratory thus takes the form of a game, with the theorist on one side, and the experimentalist and Nature on the other: the theorist is repeatedly challenged to predict the behaviour of the state with respect to the next measurement that Nature allows the experimentalist to realize, with the opportunity to refine the hypothesis as more measurement data become available.

It is thus desirable to design learning algorithms that work in the more stringent *online learning* model. Here the learner is presented a sequence of input points, say $x_1, x_2, \ldots$, one at a time. Crucially, there is no assumption whatsoever about the $x_t$'s: the sequence could be chosen adversarially, and even adaptively, which means that the choice of $x_t$ might depend on the learner's behavior on $x_1, \ldots, x_{t-1}$. The learner is trying to learn some unknown function $f(x)$, about which it initially knows only that $f$ belongs to some hypothesis class $\mathcal{H}$—or perhaps not even that; we also consider the scenario where the learner simply tries to compete with the best predictor in $\mathcal{H}$, which might or might not be a good predictor. The learning proceeds as follows: for each $t$, the learner first guesses a value $y_t$ for $f(x_t)$, and is then told the true value $f(x_t)$, or perhaps only an approximation of this value. Our goal is to design a learning algorithm with the following guarantee: *regardless of the sequence of $x_t$'s, the learner's guess, $y_t$, will be far from the true value $f(x_t)$ at most $k$ times* (where $k$, of course, is as small as possible). The $x_t$'s on which the learner errs could be spaced arbitrarily; all we require is that they be bounded in number.

This leads to the following question: can the learning theorem established by Aaronson [2007] be generalized to the online learning setting? In other words: is it true that, given a sequence $E_1, E_2, \ldots$ of yes/no measurements, where each $E_t$ is followed shortly afterward by an approximation of $\mathrm{Tr}(E_t\rho)$, there is a way to anticipate the $\mathrm{Tr}(E_t\rho)$ values by guesses $y_t \in [0,1]$, in such a way that $|y_t - \mathrm{Tr}(E_t\rho)| > \varepsilon$ at most, say, $\mathrm{O}(n)$ times (where $\varepsilon > 0$ is some constant, and $n$ again is the number of qubits)? The purpose of this paper is to provide an affirmative answer.

Throughout the paper, we consider only two-outcome measurements of an $n$ qubit mixed state $\rho$, and we specify such a measurement by a $2^n \times 2^n$ Hermitian matrix $E$ with eigenvalues in $[0,1]$. We say that $E$ "accepts" $\rho$ with probability $\mathrm{Tr}(E\rho)$ and "rejects" $\rho$ with probability $1 - \mathrm{Tr}(E\rho)$. We prove that:

**Theorem 1.** *Let $\rho$ be an $n$-qubit mixed state, and let $E_1, E_2, \ldots$ be a sequence of $2$-outcome measurements that are revealed to the learner one by one, each followed by a value $b_t \in [0, 1]$ such that $|\mathrm{Tr}(E_t \rho) - b_t| \leq \varepsilon/3$. Then there is an explicit strategy for outputting hypothesis states $\omega_1, \omega_2, \ldots$ such that $|\mathrm{Tr}(E_t \omega_t) - \mathrm{Tr}(E_t \rho)| > \varepsilon$ for at most $\mathrm{O}\!\left(\frac{n}{\varepsilon^2}\right)$ values of $t$.*

We also prove a theorem for the so-called *regret minimization model* (i.e., the "non-realizable case"), where we make no assumption about the input data arising from an actual quantum state, and our goal is simply to do not much worse than the best hypothesis state that could be found with perfect foresight. In this model, the measurements $E_1, E_2, \ldots$ are presented to a learner one-by-one. In iteration $t$, after seeing $E_t$, the learner is challenged to output a hypothesis state $\omega_t$, and then suffers a "loss" equal to $\ell_t(\mathrm{Tr}(E_t \omega_t))$ where $\ell_t$ is a real function that is revealed to the learner. Important examples of loss functions are $L_1$ loss, when $\ell_t(z) := |z - b_t|$, and $L_2$ loss, when $\ell_t(z) := (z - b_t)^2$, where $b_t \in [0, 1]$. The number $b_t$ may be an approximation of $\mathrm{Tr}(E_t \rho)$ for some fixed but unknown quantum state $\rho$, but is allowed to be arbitrary in general. In particular, the pairs $(E_t, b_t)$ may not be consistent with any quantum state. Define the *regret $R_T$*, after $T$ iterations, to be the amount by which the actual loss of the learner exceeds the loss of the best single hypothesis:

$$R_T := \sum_{t=1}^{T} \ell_t(\mathrm{Tr}(E_t \omega_t)) - \min_{\varphi} \sum_{t=1}^{T} \ell_t(\mathrm{Tr}(E_t \varphi)) \ .$$

The learner's objective is to minimize regret. We show that:

**Theorem 2.** *Let $E_1, E_2, \ldots$ be a sequence of two-outcome measurements on an $n$-qubit state presented to the learner, and $\ell_1, \ell_2, \ldots$ be the corresponding loss functions revealed in successive iterations in the regret minimization model. Suppose $\ell_t$ is convex and $L$-Lipschitz; in particular, for every $x \in \mathbb{R}$, there is a sub-derivative $\ell'_t(x)$ such that $|\ell'_t(x)| \leq L$. Then there is an explicit learning strategy that guarantees regret $R_T = \mathrm{O}(L\sqrt{Tn}\,)$ for all $T$. This is so even assuming the measurement $E_t$ and loss function $\ell_t$ are chosen adaptively, in response to the learner's previous behavior.*

*Specifically, the algorithm applies to $L_1$ loss and $L_2$ loss, and achieves regret $\mathrm{O}(\sqrt{Tn}\,)$ for both.*

The online strategies we present enjoy several advantages over full state tomography, and even over "state certification", in which we wish to test whether a given quantum state is close to a desired state or far from it. Optimal algorithms for state tomography (O'Donnell and Wright [2016], Haah et al. [2017]) or certification (Bădescu et al. [2017]) require joint measurements of an exponential number of copies of the quantum state, and assume the ability to perform noiseless, universal quantum computation. On the other hand, the algorithms implicit in Theorems 1 and 2 involve only single-copy measurements, allow for noisy measurements, and capture ground reality more closely. They produce a hypothesis state that mimics the unknown state with respect to measurements that *can be* performed in a given experimental set-up, and the accuracy of prediction improves as the set of available measurements grows. For example, in the realizable case, i.e., when the data arise from an actual quantum state, the average $L_1$ loss per iteration is $\mathrm{O}(\sqrt{n/T})$. This tends to zero, as the number of measurements becomes large. Note that $L_1$ loss may be as large as $1/2$ per iteration in the worst case, but this occurs at most $\mathrm{O}(\sqrt{nT})$ times. Finally, the algorithms have run time exponential in the number of qubits in each iteration, but are entirely *classical*. Exponential run time is unavoidable, as the measurements are presented explicitly as $2^n \times 2^n$ matrices, where $n$ is the number of qubits. If we were required to output the hypothesis states, the length of the output—also exponential in the number of qubits—would again entail exponential run time.

It is natural to wonder whether Theorems 1 and 2 leave any room for improvement. Theorem 1 is asymptotically optimal in its mistake bound of $\mathrm{O}(n/\varepsilon^2)$; this follows from the property that $n$-qubit quantum states, considered as a hypothesis class, have $\varepsilon$-fat-shattering dimension $\Theta(n/\varepsilon^2)$ (see, for example, Aaronson [2007]). On the other hand, there is room to improve Theorem 2. The bounds of which we are aware are $\Omega(\sqrt{Tn}\,)$ for the $L_1$ loss (see, e.g., [Arora et al., 2012, Theorem 4.1]) in the non-realizable case and $\Omega(n)$ for the $L_2$ loss in the realizable case, when the feedback consists of the measurement outcomes. (The latter bound, as well as an $\Omega(\sqrt{Tn}\,)$ bound for $L_1$ loss in the same setting, come from considering quantum mixed states that consist of $n$ independent classical coins, each of which could land heads with probability either $1/2$ or $1/2 + \varepsilon$. The paramater $\varepsilon$ is set to $\sqrt{n/T}$.)

We mention an application of Theorem 1, that appears in simultaneous work. Aaronson [2018] has given an algorithm for the so-called *shadow tomography* problem. Here we have an unknown $D$-dimensional pure state $\rho$, as well as known two-outcome measurements $E_1, \ldots, E_m$. Our goal is to approximate $\mathrm{Tr}(E_i \rho)$, for *every* $i$, to within additive error $\varepsilon$. We would like to do this by measuring $\rho^{\otimes k}$, where $k$ is as small as possible. Surprisingly, Aaronson [2018] showed that this can be achieved with $k = \widetilde{\mathrm{O}}((\log M)^4 (\log D)/\varepsilon^5)$, that is, a number of copies of $\rho$ that is only *polylogarithmic* in both $D$ and $M$. One component of his algorithm is essentially tantamount to online learning with $\widetilde{\mathrm{O}}(n/\varepsilon^3)$ mistakes—i.e., the learning algorithm we present in Section 4 of this paper. However, by using Theorem 1 from this paper in a black-box manner, we can improve the sample complexity of shadow tomography to $\widetilde{\mathrm{O}}((\log M)^4 (\log D)/\varepsilon^4)$. Details appear in (Aaronson [2018]).

To maximize insight, in this paper we give *three* very different approaches to proving Theorems 1 and 2 (although we do not prove every statement with all three approaches). Our first approach is to adapt techniques from online convex optimization to the setting of density matrices, which in general may be over a complex Hilbert space. This requires extending standard techniques to cope with convexity and Taylor approximations, which are widely used for functions over the real domain, but not over the complex domain. We also give an efficient iterative algorithm to produce predictions. This approach connects our problem to the modern mainstream of online learning algorithms, and achieves the best parameters (as stated in Theorems 1 and 2).

Our second approach is via a postselection-based learning procedure, which starts with the maximally mixed state as a hypothesis and then repeatedly refines it by simulating postselected measurements. This approach builds on earlier work due to Aaronson [2005], specifically the proof of $\mathsf{BQP}/\mathsf{qpoly} \subseteq \mathsf{PP}/\mathsf{poly}$. The advantage is that it is almost entirely self-contained, requiring no "power tools" from convex optimization or learning theory. On the other hand, the approach does not give optimal parameters, and we do not know how to prove Theorem 2 with it.

Our third approach is via an upper-bound on the so-called *sequential fat-shattering dimension* of quantum states, considered as a hypothesis class (see, e.g., Rakhlin et al. [2015]). In the original quantum PAC-learning theorem by Aaronson, the key step was to upper-bound the so-called *$\varepsilon$-fat-shattering dimension* of quantum states considered as a hypothesis class. Fat-shattering dimension is a real-valued generalization of VC dimension. One can then appeal to known results to get a sample-efficient learning algorithm. For online learning, however, bounding the fat-shattering dimension no longer suffices; one instead needs to consider a possibly-larger quantity called sequential fat-shattering dimension. However, by appealing to a lower bound due to Nayak [1999], Ambainis et al. [2002] for a variant of quantum *random access codes*, we are able to upper-bound the sequential fat-shattering dimension of quantum states. Using known results—in particular, those due to Rakhlin et al. [2015]—this implies the regret bound in Theorem 2, up to a multiplicative factor of $\log^{3/2} T$. The statement that *the hypothesis class of $n$-qubit states has $\varepsilon$-sequential fat-shattering dimension* $\mathrm{O}(n/\varepsilon^2)$ might be of independent interest: among other things, it implies that *any* online learning algorithm that works given bounded sequential fat-shattering dimension, will work for online learning of quantum states. We also give an alternative proof for the lower bound due to Nayak for quantum random access codes, and extend it to codes that are decoded by what we call *measurement decision trees*. We expect these also to be of independent interest.

## 1.1 Structure of the paper

We start by describing background and the technical learning setting as well as notations used throughout (Section 2). In Section 3 we give the algorithms and main theorems derived using convexity arguments and online convex optimization. In Section 4 we state the main theorem using a postselection algorithm. In Section 5 we give a sequential fat-shattering dimension bound for quantum states and its implication for online learning of quantum states. Proofs of the theorems and related claims are presented in the appendices.

## 2 Preliminaries and definitions

We define the trace norm of a matrix $M$ as $\|M\|_{\mathrm{Tr}} := \mathrm{Tr}\sqrt{MM^\dagger}$, where $M^\dagger$ is the adjoint of $M$. We denote the $i$th eigenvalue of a Hermitian matrix $X$ by $\lambda_i(X)$, its minimum eigenvalue by $\lambda_{\min}(X)$, and its maximum eigenvalue by $\lambda_{\max}(X)$. We sometimes use the notation $X \bullet Y$ to denote the trace

inner-product $\mathrm{Tr}(X^{\dagger}Y)$ between two complex matrices of the same dimensions. By 'log' we denote the natural logarithm, unless the base is explicitly mentioned.

An $n$-qubit quantum state $\rho$ is an element of $C_n$, where $C_n$ is the set of all trace-1 positive semi-definite (PSD) complex matrices of dimension $2^n$:

$$C_n = \{M \in \mathbb{C}^{2^n \times 2^n} \,,\; M = M^{\dagger} \,,\; M \succeq 0 \,,\; \mathrm{Tr}(M) = 1\} \;.$$

Note that $C_n$ is a convex set. A two-outcome measurement of an $n$-qubit state is defined by a $2^n \times 2^n$ Hermitian matrix $E$ with eigenvalues in $[0, 1]$. The measurement $E$ "accepts" $\rho$ with probability $\mathrm{Tr}(E\rho)$, and "rejects" with probability $1 - \mathrm{Tr}(E\rho)$. For the algorithms we present in this article, we assume that a two-outcome measurement is specified via a classical description of its defining matrix $E$. In the rest of the article, unless mentioned otherwise, a "measurement" refers to a "two-outcome measurement". We refer the reader to the book by Watrous [2018] for a more thorough introduction to the relevant concepts from quantum information.

**Online learning and regret.** In online learning of quantum states, we have a sequence of iterations $t = 1, 2, 3, \ldots$ of the following form. First, the learner constructs a state $\omega_t \in C_n$; we say that the learner "predicts" $\omega_t$. It then suffers a "loss" $\ell_t(\mathrm{Tr}(E_t\omega_t))$ that depends on a measurement $E_t$, both of which are presented by an adversary. Commonly used loss functions are $L_2$ loss (also called "mean square error"), given by

$$\ell_t(z) := (z - b_t)^2 \;,$$

and $L_1$ loss (also called "absolute loss"), given by

$$\ell_t(z) := |z - b_t| \;,$$

where $b_t \in [0, 1]$. The parameter $b_t$ may be an approximation of $\mathrm{Tr}(E_t\rho)$ for some fixed quantum state $\rho$ not known to the learner, obtained by measuring multiple copies of $\rho$. However, in general, the parameter is allowed to be arbitrary.

The learner then "observes" feedback from the measurement $E_t$; the feedback is also provided by the adversary. The simplest feedback is the realization of a binary random variable $Y_t$ such that

$$Y_t = \begin{cases} 1 & \text{with probability } \mathrm{Tr}(E_t\rho) \;, \qquad \text{and} \\ 0 & \text{with probability } 1 - \mathrm{Tr}(E_t\rho) \;. \end{cases}$$

Another common feedback is a number $b_t$ as described above, especially in case that the learner suffers $L_1$ or $L_2$ loss.

We would like to design a strategy for updating $\omega_t$ based on the loss, measurements, and feedback in all the iterations so far, so that the learner's total loss is minimized in the following sense. We would like that over $T$ iterations (for a number $T$ known in advance), the learner's total loss is not much more than that of the hypothetical strategy of outputting the same quantum state $\varphi$ at every time step, where $\varphi$ minimizes the total loss *with perfect hindsight*. Formally this is captured by the notion of *regret* $R_T$, defined as

$$R_T := \sum_{t=1}^{T} \ell_t(\mathrm{Tr}(E_t\omega_t)) - \min_{\varphi \in C_n} \sum_{t=1}^{T} \ell_t(\mathrm{Tr}(E_t\varphi)) \;.$$

The sequence of measurements $E_t$ can be arbitrary, even adversarial, based on the learner's previous actions. Note that if the loss function is given by a fixed state $\rho$ (as in the case of mean square error), the minimum total loss would be 0. This is called the "realizable" case. However, in general, the loss function presented by the adversary need not be consistent with any quantum state. This is called the "non-realizable" case.

A special case of the online learning setting is called *agnostic learning*; here the measurements $E_t$ are drawn from a fixed and unknown distribution $\mathcal{D}$. The setting is called "agnostic" because we still do not assume that the losses correspond to any actual state $\rho$ (i.e., the setting may be non-realizable).

**Online mistake bounds.** In some online learning scenarios the quantity of interest is not the mean square error, or some other convex loss, but rather simply the total number of "mistakes" made. For example, we may be interested in the number of iterations in which the predicted probability of

acceptance $\text{Tr}(E_t\omega_t)$ is more than $\varepsilon$-far from the actual value $\text{Tr}(E_t\rho)$, where $\rho$ is again a fixed state not known to the learner. More formally, let

$$\ell_t(\text{Tr}(E_t\omega_t)) := |\text{Tr}(E_t\omega_t) - \text{Tr}(E_t\rho)|$$

be the absolute loss function. Then the goal is to bound the number of iterations in which $\ell_t(\text{Tr}(E_t\omega_t)) > \varepsilon$, regardless of the sequence of measurements $E_t$ presented by the adversary. We assume that in this setting, the adversary provides as feedback an approximation $b_t \in [0,1]$ that satisfies $|\text{Tr}(E_t\rho) - b_t| \leq \frac{\varepsilon}{3}$.

## 3 Online learning of quantum states

In this section, we use techniques from online convex optimization to minimize regret. The same algorithms may be adapted to also minimize the number of mistakes made.

### 3.1 Regularized Follow-the-Leader

We first follow the template of the Regularized Follow-the-Leader algorithm (RFTL; see, for example, [Hazan, 2015, Chapter 5]). The algorithm below makes use of von Neumann entropy, which relates to the Matrix Exponentiated Gradient algorithm (Tsuda et al. [2005]).

---

**Algorithm 1** RFTL for Quantum Tomography

---

1: Input: $T, \mathcal{K} := C_n, \eta < \frac{1}{2}$
2: Set $\omega_1 := 2^{-n}\mathbb{I}$.
3: **for** $t = 1, \ldots, T$ **do**
4:     Predict $\omega_t$. Consider the convex and $L$-Lipschitz loss function $\ell_t : \mathbb{R} \to \mathbb{R}$ given by measurement $E_t : \ell_t(\text{Tr}(E_t\varphi))$. Let $\ell'_t(x)$ be a sub-derivative of $\ell_t$ with respect to $x$. Define

$$\nabla_t := \ell'_t(\text{Tr}(E_t\omega_t))E_t \ .$$

5:     Update decision according to the RFTL rule with von Neumann entropy:

$$\omega_{t+1} := \underset{\varphi \in \mathcal{K}}{\arg\min} \left\{ \eta \sum_{s=1}^{t} \text{Tr}(\nabla_s\varphi) + \sum_{i=1}^{2^n} \lambda_i(\varphi) \log \lambda_i(\varphi) \right\} \ . \tag{1}$$

6: **end for**

---

**Remark 1:** The mathematical program in Eq. (1) is convex, and thus can be solved in polynomial time in the dimension, which is $2^n$.

**Theorem 3.** *Setting $\eta = \sqrt{\frac{(\log 2)n}{2TL^2}}$, the regret of Algorithm 1 is bounded by $2L\sqrt{(2\log 2)Tn}$.*

**Remark 2:** In the case where the feedback is an independent random variable $Y_t$, where $Y_t = 0$ with probability $1 - \text{Tr}(E_t\rho)$ and $Y_t = 1$ with probability $\text{Tr}(E_t\rho)$ for a fixed but unknown state $\rho$, we define $\nabla_t$ in Algorithm 1 as $\nabla_t := 2(\text{Tr}(E_t\omega_t) - Y_t)E_t$. Then $\mathbb{E}[\nabla_t]$ is the gradient of the $L_2$ loss function where we receive precise feedback $\text{Tr}(E_t\rho)$ instead of $Y_t$. It follows from the proof of Theorem 3 that the expected $L_2$ regret of Algorithm 1, namely

$$\mathbb{E}\left[ \sum_{t=1}^{T} (\text{Tr}(E_t\omega_t) - \text{Tr}(E_t\rho))^2 \right] \ ,$$

is bounded by $\text{O}(\sqrt{Tn})$.

The proof of Theorem 3 appears in Appendix B. The proof is along the lines of [Hazan, 2015, Theorem 5.2], except that the loss function does not take a raw state as input, and our domain for optimization is complex. Therefore, the mean value theorem does not hold, which means we need to approximate the Bregman divergence instead of replacing it by a norm as in the original proof. Another subtlety is that convexity needs to be carefully defined with respect to the complex domain.

## 3.2 Matrix Multiplicative Weights

The Matrix Multiplicative Weights (MMW) algorithm [Arora and Kale, 2016] provides an alternative means of proving Theorem 2. The algorithm follows the template of Algorithm 1 with step 5 replaced by the following update rule:

$$\omega_{t+1} := \frac{\exp(-\frac{\eta}{L} \sum_{\tau=1}^{t} \nabla_\tau)}{\text{Tr}(\exp(-\frac{\eta}{L} \sum_{\tau=1}^{t} \nabla_\tau))} \quad . \tag{2}$$

In the notation of Arora and Kale [2016], this algorithm is derived using the loss matrices $M_t = \frac{1}{L}\nabla_t = \frac{1}{L}\ell'_t(\text{Tr}(E_t\omega_t))E_t$. Since $\|E_t\| \leq 1$ and $|\ell'_t(\text{Tr}(E_t\omega_t))| \leq L$, we have $\|M_t\| \leq 1$, as required in the analysis of the Matrix Multiplicative Weights algorithm. We have the following regret bound for the algorithm (proved in Appendix C):

**Theorem 4.** *Setting* $\eta = \sqrt{\frac{(\log 2)n}{4T}}$, *the regret of the algorithm based on the update rule* (2) *is bounded by* $2L\sqrt{(\log 2)Tn}$.

## 3.3 Proof of Theorem 1

Consider either the RFTL or MMW based online learning algorithm described in the previous subsections, with the 1-Lipschitz convex absolute loss function $\ell_t(x) = |x - b_t|$. We run the algorithm in a sub-sequence of the iterations, using only the measurements presented in those iterations. The subsequence of iterations is determined as follows. Let $\omega_t$ denote the hypothesis maintained by the algorithm in iteration $t$. We run the algorithm in iteration $t$ if $\ell_t(\text{Tr}(E_t\omega_t)) > \frac{2\varepsilon}{3}$. Note that whenever $|\text{Tr}(E_t\omega_t) - \text{Tr}(E_t\rho)| > \varepsilon$, we have $\ell_t(\text{Tr}(E_t\omega_t)) > \frac{2\varepsilon}{3}$, so we update the hypothesis according to the RFTL/MMW rule in that iteration.

As we explain next, the algorithm makes at most $\text{O}(\frac{n}{\varepsilon^2})$ updates regardless of the number of measurements presented (i.e., regardless of the number of iterations), giving the required mistake bound. For the true quantum state $\rho$, we have $\ell_t(\text{Tr}(E_t\rho)) < \frac{\varepsilon}{3}$ for all $t$. Thus if the algorithm makes $T$ updates (i.e., we run the algorithm in $T$ of the iterations), the regret bound implies that $\frac{2\varepsilon}{3}T \leq \frac{\varepsilon}{3}T + \text{O}(\sqrt{Tn})$. Simplifying, we get the bound $T = \text{O}(\frac{n}{\varepsilon^2})$, as required.

# 4 Learning Using Postselection

In this section, we give a direct route to proving a slightly weaker version of Theorem 1: one that does not need the tools of convex optimization, but only tools intrinsic to quantum information.

In the following, by a "register" we mean a designated sequence of qubits. Given a two-outcome measurement $E$ on $n$-qubits states, we define an operator $\mathcal{M}$ that "postselects" on acceptance by $E$. (While a measurement results in a random outcome distributed according to the probability of acceptance or rejection, *postselection* is a hypothetical operation that produces an outcome of one's choice with certainty.) Let $U$ be any unitary operation on $n + 1$ qubits that maps states of the form $|\psi\rangle|0\rangle$ to $\sqrt{E}\,|\psi\rangle|0\rangle + \sqrt{\mathbb{I} - E}\,|\psi\rangle|1\rangle$. Such a unitary operation always exists (see, e.g., [Watrous, 2018, Theorem 2.42]). Denote the $(n+1)$th qubit by register $B$. Let $\Pi := \mathbb{I} \otimes |0\rangle\langle0|$ be the orthogonal projection onto states that equal $|0\rangle$ in register $B$. Then we define the operator $\mathcal{M}$ as

$$\mathcal{M}(\varphi) := \frac{1}{\text{Tr}(E\varphi)}\,\text{Tr}_B\big(U^{-1}\Pi U\,(\varphi \otimes |0\rangle\langle0|)\,U^{-1}\Pi U\big) \quad , \tag{3}$$

if $\text{Tr}(E\varphi) \neq 0$, and $\mathcal{M}(\varphi) := 0$ otherwise. Here, $\text{Tr}_B$ is the *partial trace* operator over qubit $B$ [Watrous, 2018, Section 1.1]. This operator $\mathcal{M}$ has the effect of mapping the quantum state $\varphi$ to the (normalized) post-measurement state when we perform the measurement $E$ and get outcome "yes" (i.e., the measurement "accepts"). We emphasize that we use a fresh ancilla qubit initialized to state $|0\rangle$ as register $B$ in every application of the operator $\mathcal{M}$. We say that the postselection succeeds with probability $\text{Tr}(E\varphi)$.

We need a slight variant of a well-known result, which Aaronson called the "Quantum Union Bound" (see, for example, Aaronson [2006, 2016], Wilde [2013]).

**Theorem 5** (variant of Quantum Union Bound; Gao [2015]). *Suppose we have a sequence of two-outcome measurements $E_1, \ldots, E_k$, such that each $E_i$ accepts a certain mixed state $\varphi$ with probability at least $1-\varepsilon$. Consider the corresponding operators $\mathcal{M}_1, \mathcal{M}_2, \ldots, \mathcal{M}_k$ that postselect on acceptance by the respective measurements $E_1, E_2, \ldots, E_k$. Let $\widetilde{\varphi}$ denote the state $(\mathcal{M}_k \mathcal{M}_{k-1} \cdots \mathcal{M}_1)(\varphi)$ obtained by applying each of the $k$ postselection operations in succession. Then the probability that all the postselection operations succeed, i.e., the $k$ measurements all accept $\varphi$, is at least $1 - 2\sqrt{k\varepsilon}$. Moreover, $\|\widetilde{\varphi} - \varphi\|_{\mathrm{Tr}} \leq 4\sqrt{k\varepsilon}$.*

We may infer the above theorem by applying Theorem 1 from (Gao [2015]) to the state $\varphi$ augmented with $k$ ancillary qubits $B_1, B_2, \ldots, B_k$ initialized to 0, and considering $k$ orthogonal projection operators $U_i^{-1} \Pi_i U_i$, where the unitary operator $U_i$ and the projection operator $\Pi_i$ are as defined for the postselection operation $\mathcal{M}_i$ for $E_i$. The $i$th projection operator $U_i^{-1} \Pi_i U_i$ acts on the registers holding $\varphi$ and the $i$th ancillary qubit $B_i$.

We prove the main result of this section using suitably defined postselection operators in an online learning algorithm (proof in Appendix D):

**Theorem 6.** *Let $\rho$ be an unknown $n$-qubit mixed state, let $E_1, E_2, \ldots$ be a sequence of two-outcome measurements, and let $\varepsilon > 0$. There exists a strategy for outputting hypothesis states $\omega_0, \omega_1, \ldots,$ where $\omega_t$ depends only on $E_1, \ldots, E_t$ and real numbers $b_1, \ldots, b_t$ in $[0, 1]$, such that as long as $|b_t - \mathrm{Tr}(E_t \rho)| \leq \varepsilon/3$ for every $t$, we have*

$$|\mathrm{Tr}(E_{t+1} \omega_t) - \mathrm{Tr}(E_{t+1} \rho)| > \varepsilon$$

*for at most $\mathrm{O}\!\left(\frac{n}{\varepsilon^3} \log \frac{n}{\varepsilon}\right)$ values of $t$. Here the $E_t$'s and $b_t$'s can otherwise be chosen adversarially.*

## 5 Learning Using Sequential Fat-Shattering Dimension

In this section, we prove regret bounds using the notion of *sequential fat-shattering dimension*. Let $S$ be a set of functions $f : U \to [0, 1]$, and $\varepsilon > 0$. Then, following Rakhlin et al. [2015], let the $\varepsilon$-*sequential fat-shattering dimension* of $S$, or $\mathrm{sfat}_\varepsilon(S)$, be the largest $k$ for which we can construct a complete binary tree $T$ of depth $k$, such that

- each internal vertex $v \in T$ has associated with it a point $x_v \in U$ and a real $a_v \in [0, 1]$, and
- for each leaf vertex $v \in T$ there exists an $f \in S$ that causes us to reach $v$ if we traverse $T$ from the root such that at any internal node $w$ we traverse the left subtree if $f(x_w) \leq a_w - \varepsilon$ and the right subtree if $f(x_w) \geq a_w + \varepsilon$. If we view the leaf $v$ as a $k$-bit string, the function $f$ is such that for all ancestors $u$ of $v$, we have $f(x_u) \leq a_u - \varepsilon$ if $v_i = 0$, and $f(x_u) \geq a_u + \varepsilon$ if $v_i = 1$, when $u$ is at depth $i - 1$ from the root.

An $n$-qubit state $\rho$ induces a function $f$ on the set of two-outcome measurements $E$ defined as $f(E) := \mathrm{Tr}(E\rho)$. With this correspondence in mind, we establish a bound on the sequential fat-shattering dimension of the set of $n$-qubit quantum states. The bound is based on a generalization of "random access coding" (Nayak [1999], Ambainis et al. [2002]) called "serial encoding". We derive the following bound on the length of serial encoding. Let $\mathrm{H}(x) := -x \log_2 x - (1-x) \log_2 (1-x)$ be the binary entropy function.

**Corollary 7.** *Let $k$ and $n$ be positive integers. For each $k$-bit string $y := y_1 \cdots y_k$, let $\rho_y$ be an $n$-qubit mixed state such that for each $i \in \{1, 2, \ldots, k\}$, there is a two-outcome measurement $E'$ that depends only on $i$ and the prefix $v := y_1 y_2 \cdots y_{i-1}$, and has the following properties*

*(iii) if $y_i = 0$ then $\mathrm{Tr}(E' \rho_y) \leq a_v - \varepsilon$, and*

*(iv) if $y_i = 1$ then $\mathrm{Tr}(E' \rho_y) \geq a_v + \varepsilon$,*

*where $\varepsilon \in (0, 1/2]$ and $a_v \in [0, 1]$ is a "pivot point" associated with the prefix $v$. Then*

$$n \;\geq\; \left(1 - \mathrm{H}\!\left(\frac{1 - \varepsilon}{2}\right)\right) k \;.$$

*In particular, $k = \mathrm{O}\!\left(n/\varepsilon^2\right)$.*

(The proof is presented in Appendix E).

Corollary 7 immediately implies the following theorem:

**Theorem 8.** *Let $U$ be the set of two-outcome measurements $E$ on an $n$-qubit state, and let $S$ be the set of all functions $f : U \to [0,1]$ that have the form $f(E) := \mathrm{Tr}(E\rho)$ for some $\rho$. Then for all $\varepsilon > 0$, we have $\mathrm{sfat}_\varepsilon(S) = \mathrm{O}\big(n/\varepsilon^2\big)$.*

Theorem 8 strengthens an earlier result due to Aaronson [2007], which proved the same upper bound for the "ordinary" (non-sequential) fat-shattering dimension of quantum states considered as a hypothesis class.

Now we may use existing results from the literature, which relate sequential fat-shattering dimension to online learnability. In particular, in the non-realizable case, Rakhlin et al. [2015] recently showed the following:

**Theorem 9** (Rakhlin et al. [2015]). *Let $S$ be a set of functions $f : U \to [0,1]$ and for every integer $t \geq 1$, let $\ell_t : [0,1] \to \mathbb{R}$ be a convex, $L$-Lipschitz loss function. Suppose we are sequentially presented elements $x_1, x_2, \ldots \in U$, with each $x_t$ followed by the loss function $\ell_t$. Then there exists a learning strategy that lets us output a sequence of hypotheses $f_1, f_2, \ldots \in S$, such that the regret is upper-bounded as:*

$$\sum_{t=1}^{T} \ell_t \left( f_t(x_t) \right) \leq \min_{f \in S} \sum_{t=1}^{T} \ell_t \left( f(x_t) \right) + 2LT \inf_{\alpha} \left\{ 4\alpha + \frac{12}{\sqrt{T}} \int_{\alpha}^{1} \sqrt{\mathrm{sfat}_\beta(S) \log \left( \frac{2\mathrm{e}T}{\beta} \right)} \mathrm{d}\beta \right\}.$$

This follows from Theorem 8 in (Rakhlin et al. [2015]) as in the proof of Proposition 9 in the same article.

Combining Theorem 8 with Theorem 9 gives us the following:

**Corollary 10.** *Suppose we are presented with a sequence of two-outcome measurements $E_1, E_2, \ldots$ of an $n$-qubit state, with each $E_t$ followed by a loss function $\ell_t$ as in Theorem 9. Then there exists a learning strategy that lets us output a sequence of hypothesis states $\omega_1, \omega_2, \ldots$ such that the regret after the first $T$ iterations is upper-bounded as:*

$$\sum_{t=1}^{T} \ell_t \left( \mathrm{Tr}(E_t\omega_t) \right) \leq \min_{\omega \in C_n} \sum_{t=1}^{T} \ell_t \left( \mathrm{Tr}(E_t\omega) \right) + \mathrm{O}\left( L\sqrt{nT} \log^{3/2} T \right).$$

Note that the result due to Rakhlin et al. [2015] is non-explicit. In other words, by following this approach, we do not derive any specific online learning algorithm for quantum states that has the stated upper bound on regret; we only prove non-constructively that such an algorithm exists.

We expect that the approach in this section, based on sequential fat-shattering dimension, could also be used to prove a mistake bound for the realizable case, but we leave that to future work.

## 6   Open Problems

We conclude with some questions arising from this work. The regret bound established in Theorem 2 for $L_1$ loss is tight. Can we similarly achieve optimal regret for other loss functions of interest, for example for $L_2$-loss? It would also be interesting to obtain regret bounds in terms of the loss of the best quantum state in hindsight, as opposed to $T$ (the number of iterations), using the techniques in this article. Such a bound has been shown by [Tsuda et al., 2005, Lemma 3.2] for $L_2$-loss using the Matrix Exponentiated Gradient method.

In what cases can one do online learning of quantum states, not only with few samples, but also with a polynomial amount of computation? What is the tight generalization of our results to measurements with $d$ outcomes? Is it the case, in online learning of quantum states, that *any* algorithm works, so long as it produces hypothesis states that are approximately consistent with all the data seen so far? Note that none of our three proof techniques seem to imply this general conclusion.

## Footnotes

*Supported by a Vannevar Bush Faculty Fellowship from the US Department of Defense. Part of this work was done while the author was supported by an NSF Alan T. Waterman Award.

†Part of this work was done when the author was a research assistant at Princeton University.

‡Research supported in part by NSERC Canada.

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
