[Supplementary Material · supplement_material_1.pdf]

## A  Auxiliary Lemmas

The following lemma is from (Tsuda et al. [2005]), given here for completeness.

**Lemma 11.** *For Hermitian matrices $A, B$ and Hermitian PSD matrix $X$, if $A \succeq B$, then $\operatorname{Tr}(AX) \geq \operatorname{Tr}(BX)$.*

*Proof.* Let $C := A - B$. By definition, $C \succeq 0$. It suffices to show that $\operatorname{Tr}(CX) \geq 0$. Let $VQV^\dagger$ be the eigen-decomposition of $X$, and let $C = VPV^\dagger$, where $P := V^\dagger C V \succeq 0$. Then $\operatorname{Tr}(CX) = \operatorname{Tr}(VPQV^\dagger) = \operatorname{Tr}(PQ) = \sum_{i=1}^n P_{ii}Q_{ii}$. Since $P \succeq 0$ and all the eigenvalues of $X$ are nonnegative, $P_{ii} \geq 0, Q_{ii} \geq 0$. Therefore $\operatorname{Tr}(CX) \geq 0$. $\qquad\square$

**Lemma 12.** *If $A, B$ are Hermitian matrices, then $\operatorname{Tr}(AB) \in \mathbb{R}$.*

*Proof.* The proof is similar to Lemma 11. Let $VQV^\dagger$ be the eigendecomposition of $A$. Then $Q$ is a real diagonal matrix. We have $B = VPV^\dagger$, where $P := V^\dagger B V$. Note that $P^\dagger = V^\dagger B^\dagger V = P$, so $P$ has a real diagonal. Then $\operatorname{Tr}(AB) = \operatorname{Tr}(VQV^\dagger VPV^\dagger) = \operatorname{Tr}(VQPV^\dagger) = \operatorname{Tr}(QP) = \sum_{i=1}^n Q_{ii}P_{ii}$. Since $Q_{ii}, P_{ii} \in \mathbb{R}$ for all $i$, $\operatorname{Tr}(AB) \in \mathbb{R}$. $\qquad\square$

## B  Proof of Theorem 3

*Proof of Theorem 3.* Since $\ell_t$ is convex, for all $\varphi \in \mathcal{K}$,

$$\ell_t(\operatorname{Tr}(E_t\omega_t)) - \ell_t(\operatorname{Tr}(E_t\varphi)) \leq \ell_t'(\operatorname{Tr}(E_t\omega_t))\left[\operatorname{Tr}(E_t\omega_t) - \operatorname{Tr}(E_t\varphi)\right] = \nabla_t \bullet (\omega_t - \varphi) \ .$$

(Recall that '$\bullet$' denotes the trace inner-product between complex matrices of the same dimensions.) Summing over $t$,

$$\sum_{t=1}^T [\ell_t(\operatorname{Tr}(E_t\omega_t)) - \ell_t(\operatorname{Tr}(E_t\varphi))] \leq \sum_{t=1}^T \left[\operatorname{Tr}(\nabla_t\omega_t) - \operatorname{Tr}(\nabla_t\varphi)\right] \ .$$

Define $g_t(X) = \nabla_t \bullet X$, and $g_0(X) = \frac{1}{\eta}R(X)$, where $R(X)$ is the negative von Neumann Entropy of $X$ (in nats). Denote $D_R^2 := \max_{\varphi,\varphi' \in \mathcal{K}}\{R(\varphi) - R(\varphi')\}$. By [Hazan, 2015, Lemma 5.2], for any $\varphi \in \mathcal{K}$, we have

$$\sum_{t=1}^T [g_t(\omega_t) - g_t(\varphi)] \leq \sum_{t=1}^T \nabla_t \bullet (\omega_t - \omega_{t+1}) + \frac{1}{\eta}D_R^2 \ . \tag{4}$$

Define $\Phi_t(X) = \{\eta \sum_{s=1}^t \nabla_s \bullet X + R(X)\}$, then the convex program in line 5 of Algorithm 1 finds the minimizer of $\Phi_t(X)$ in $\mathcal{K}$. The following claim shows that that the minimizer is always *positive definite* (proof provided later in this section):

**Claim 13.** *For all $t \in \{1, 2, ..., T\}$, we have $\omega_t \succ 0$.*

For $X \succ 0$, we can write $R(X) = \operatorname{Tr}(X \log X)$, and define

$$\nabla\Phi_t(X) := \eta \sum_{s=1}^t \nabla_s + \mathbb{I} + \log X \ .$$

The definition of $\nabla\Phi_t(X)$ is analogous to the gradient of $\Phi_t(X)$ if the function is defined over real symmetric matrices. Moreover, the following condition, similar to the optimality condition over a real domain, is satisfied (proof provided later in this section).

**Claim 14.** *For all $t \in \{1, 2, \ldots, T-1\}$,*

$$\nabla\Phi_t(\omega_{t+1}) \bullet (\omega_t - \omega_{t+1}) \geq 0 \ . \tag{5}$$

Denote
$$B_{\Phi_t}(\omega_t\|\omega_{t+1}) := \Phi_t(\omega_t) - \Phi_t(\omega_{t+1}) - \nabla\Phi_t(\omega_{t+1}) \bullet (\omega_t - \omega_{t+1}) \ .$$
Then by the Pinsker inequality (see, for example, Carlen and Lieb [2014] and the references therein),
$$\frac{1}{2}\|\omega_t - \omega_{t+1}\|_{\mathrm{Tr}}^2 \leq \mathrm{Tr}(\omega_t \log \omega_t) - \mathrm{Tr}(\omega_t \log \omega_{t+1}) = B_{\Phi_t}(\omega_t\|\omega_{t+1}) \ .$$

We have
$$\begin{aligned}
B_{\Phi_t}(\omega_t\|\omega_{t+1}) &= \Phi_t(\omega_t) - \Phi_t(\omega_{t+1}) - \nabla\Phi_t(\omega_{t+1}) \bullet (\omega_t - \omega_{t+1})\\
&\leq \Phi_t(\omega_t) - \Phi_t(\omega_{t+1})\\
&= \Phi_{t-1}(\omega_t) - \Phi_{t-1}(\omega_{t+1}) + \eta\nabla_t \bullet (\omega_t - \omega_{t+1})\\
&\leq \eta\nabla_t \bullet (\omega_t - \omega_{t+1}) \ ,
\end{aligned} \tag{6}$$
where the first inequality follows from Claim 14, and the second because $\Phi_{t-1}(\omega_t) \leq \Phi_{t-1}(\omega_{t+1})$ ($\omega_t$ minimizes $\Phi_{t-1}(X)$). Therefore
$$\frac{1}{2}\|\omega_t - \omega_{t+1}\|_{\mathrm{Tr}}^2 \leq \eta\nabla_t \bullet (\omega_t - \omega_{t+1}) \ . \tag{7}$$

Let $\|M\|_{\mathrm{Tr}}^*$ denote the dual of the trace norm, i.e., the spectral norm of the matrix $M$. By Generalized Cauchy-Schwartz [Bhatia, 1997, Exercise IV.1.14, page 90],
$$\begin{aligned}
\nabla_t \bullet (\omega_t - \omega_{t+1}) &\leq \|\nabla_t\|_{\mathrm{Tr}}^* \|\omega_t - \omega_{t+1}\|_{\mathrm{Tr}}\\
&\leq \|\nabla_t\|_{\mathrm{Tr}}^* \sqrt{2\eta\nabla_t \bullet (\omega_t - \omega_{t+1})} \ . \qquad \text{by Eq. (7)}.
\end{aligned}$$
Rearranging,
$$\nabla_t \bullet (\omega_t - \omega_{t+1}) \leq 2\eta\|\nabla_t\|_{\mathrm{Tr}}^{*2} \leq 2\eta G_R^2 \ ,$$
where $G_R$ is an upper bound on $\|\nabla_t\|_{\mathrm{Tr}}^*$. Combining with Eq. (4), we arrive at the following bound
$$\sum_{t=1}^T \nabla_t \bullet (\omega_t - \varphi) \leq \sum_{t=1}^T \nabla_t \bullet (\omega_t - \omega_{t+1}) + \frac{1}{\eta}D_R^2 \leq 2\eta T G_R^2 + \frac{1}{\eta}D_R^2 \ .$$
Taking $\eta = \frac{D_R}{G_R\sqrt{2T}}$, we get $\sum_{t=1}^T \nabla_t \bullet (\omega_t - \varphi) \leq 2D_R G_R\sqrt{2T}$. Going back to the regret bound,
$$\sum_{t=1}^T [\ell_t(\mathrm{Tr}(E_t\omega_t)) - \ell_t(\mathrm{Tr}(E_t\varphi))] \leq \sum_{t=1}^T \nabla_t \bullet (\omega_t - \varphi) \leq 2D_R G_R\sqrt{2T} \ .$$

We proceed to show that $D_R = \sqrt{(\log 2)n}$. Let $\Delta_{2^n}$ denote the set of probability distributions over $[2^n]$. By definition,
$$D_R^2 = \max_{\varphi,\varphi'\in\mathcal{K}}\{R(\varphi) - R(\varphi')\} = \max_{\varphi\in\mathcal{K}} -R(\varphi) = \max_{\lambda\in\Delta_{2^n}}\sum_{i=1}^{2^n}\lambda_i \log\frac{1}{\lambda_i} = n\log 2 \ .$$

Since the dual norm of the trace norm is the spectral norm, we have
$$\|\nabla_t\|_{\mathrm{Tr}}^* = \|\ell_t'(\mathrm{Tr}(E_t\omega_t))E_t\| \leq L\|E_t\| \leq L \ .$$
Therefore $\sum_{t=1}^T [(\ell_t(\mathrm{Tr}(E_t\omega_t)) - \ell_t(\mathrm{Tr}(E_t\varphi))] \leq 2L\sqrt{(2\log 2)nT}$. $\qquad\square$

*Proof of Claim 13.* Let $P \in \mathcal{K}$ be such that $\lambda_{\min}(P) = 0$. Suppose $P = VQV^\dagger$, where $Q$ is a diagonal matrix with real values on the diagonal. Assume that $Q_{1,1} = \lambda_{\max}(P)$ and $Q_{2^n,2^n} = \lambda_{\min}(P) = 0$. Let $P' = VQ'V^\dagger$ such that $Q'_{1,1} = Q_{1,1} - \varepsilon$, $Q'_{2^n,2^n} = \varepsilon$ for $\varepsilon < \lambda_{\max}(P)$, and $Q'_{ii} = Q_{ii}$ for $i \in \{2,3,...,2^n-1\}$, so $P' \in \mathcal{K}$. We show that there exists $\varepsilon > 0$ such that $\Phi_t(P') \leq \Phi_t(P)$. Expanding both sides of the inequality, we see that it is equivalent to showing that for some $\varepsilon$,
$$\eta\sum_{s=1}^t \nabla_s \bullet (P' - P) \leq \lambda_1(P)\log\lambda_1(P) - \lambda_1(P')\log\lambda_1(P') - \varepsilon\log\varepsilon \ .$$

Let $\alpha = \lambda_1(P) = Q_{1,1}$, and $A = \eta \sum_{s=1}^{t} \nabla_s$. The inequality then becomes

$$A \bullet (P' - P) \leq \alpha \log \alpha - (\alpha - \varepsilon) \log(\alpha - \varepsilon) - \varepsilon \log \varepsilon \ .$$

Observe that $\|A\| \leq \eta \sum_{s=1}^{t} \|\nabla_s\| = \eta \sum_{s=1}^{t} \|\ell'_s(\mathrm{Tr}(E_s \omega_s)) E_s\| \leq \eta L t$. So by the Generalized Cauchy-Schwartz inequality,

$$A \bullet (P' - P) \leq \eta L t \, \|P' - P\|_{\mathrm{Tr}} \leq 2\varepsilon \eta L t \ .$$

Since $\eta, t, \alpha, L$ are finite and $-\log \varepsilon \to \infty$ as $\varepsilon \to 0$, there exists $\varepsilon$ small such that $2\eta L t \leq \log \alpha - \log \varepsilon$. We have

$$\begin{aligned}
2\eta L t \varepsilon &\leq \varepsilon \log \alpha - \varepsilon \log \varepsilon \\
&= \alpha \log \alpha - (\alpha - \varepsilon) \log \alpha - \varepsilon \log \varepsilon \\
&\leq \alpha \log \alpha - (\alpha - \varepsilon) \log(\alpha - \varepsilon) - \varepsilon \log \varepsilon \ .
\end{aligned}$$

So there exists $\varepsilon > 0$ such that $\Phi_t(P') \leq \Phi_t(P)$. If $P$ has multiple eigenvalues that are 0, we can repeat the proof and show that there exists a PD matrix $P'$ such that $\Phi_t(P') \leq \Phi_t(P)$. Since $\omega_t$ is a minimizer of $\Phi_{t-1}$ and $\omega_1 \succ 0$, we conclude that $\omega_t \succ 0$ for all $t$. ☐

*Proof of Claim 14.* Suppose $\nabla \Phi_t(\omega_{t+1}) \bullet (\omega_t - \omega_{t+1}) < 0$. Let $a \in (0,1)$ and $\bar{X} = (1-a)\omega_{t+1} + a\omega_t$, then $\bar{X}$ is a density matrix and is positive definite. Define $\triangle = \bar{X} - \omega_{t+1} = a(\omega_t - \omega_{t+1})$. We have

$$\begin{aligned}
\Phi_t(\bar{X}) - \Phi_t(\omega_{t+1}) &= a\nabla\Phi_t(\omega_{t+1}) \bullet (\omega_t - \omega_{t+1}) + B_{\Phi_t}(\bar{X}\|\omega_{t+1}) \\
&\leq a\nabla\Phi_t(\omega_{t+1}) \bullet (\omega_t - \omega_{t+1}) + \frac{\mathrm{Tr}(\triangle^2)}{\lambda_{\min}(\omega_{t+1})} \\
&= a\nabla\Phi_t(\omega_{t+1}) \bullet (\omega_t - \omega_{t+1}) + \frac{a^2 \, \mathrm{Tr}((\omega_t - \omega_{t+1})^2)}{\lambda_{\min}(\omega_{t+1})} \ .
\end{aligned}$$

The above inequality is due to [Audenaert and Eisert, 2005, Theorem 2]. Dividing by $a$ on both sides, we have

$$\frac{\Phi_t(\bar{X}) - \Phi_t(\omega_{t+1})}{a} \leq \nabla\Phi_t(\omega_{t+1}) \bullet (\omega_t - \omega_{t+1}) + \frac{a\,\mathrm{Tr}((\omega_t - \omega_{t+1})^2)}{\lambda_{\min}(\omega_{t+1})} \ .$$

So we can find $a$ small enough such that the right hand side of the above inequality is negative. However, we would have $\Phi_t(\bar{X}) - \Phi_t(\omega_{t+1}) < 0$, which is a contradiction. So $\nabla\Phi_t(\omega_{t+1}) \bullet (\omega_t - \omega_{t+1}) \geq 0$. ☐

## C  Proof of Theorem 4

*Proof of Theorem 4.* Note that for any density matrix $\varphi$, we have $M_t \bullet \varphi = \frac{1}{L}\ell'_t(\mathrm{Tr}(E_t\omega_t)) \mathrm{Tr}(E_t\varphi)$. Then, the regret bound for Matrix Multiplicative Weights [Arora and Kale, 2016, Theorem 3.1] implies that for any density matrix $\varphi$, we have

$$\sum_{t=1}^{T} \ell'_t(\mathrm{Tr}(E_t\omega_t)) \mathrm{Tr}(E_t\omega_t) \leq \sum_{t=1}^{T} \ell'_t(\mathrm{Tr}(E_t\omega_t)) \mathrm{Tr}(E_t\varphi) + \eta L T + \frac{L \log(2^n)}{\eta} \ .$$

Here, we used the bound $M_t^2 \bullet \omega_t \leq 1$. Next, since $\ell_t$ is convex, we have

$$\ell'_t(\mathrm{Tr}(E_t\omega_t)) \mathrm{Tr}(E_t\omega_t) - \ell'_t(\mathrm{Tr}(E_t\omega_t)) \mathrm{Tr}(E_t\varphi) \geq \ell_t(\mathrm{Tr}(E_t\omega_t)) - \ell_t(\mathrm{Tr}(E_t\varphi)) \ .$$

Using this bound, and the stated value of $\eta$, we get the required regret bound. ☐

## D  Proof of Theorem 6

*Proof of Theorem 6.* Let $\rho^* := \rho^{\otimes k}$ be an amplified version of $\rho$, over a Hilbert space of dimension $D := 2^{kn}$, for some $k$ to be set later. Throughout, we maintain a classical description of a $D$-dimensional "amplified hypothesis state" $\omega_t^*$, which we view as being the state of $k$ registers with $n$

qubits each. We ensure that $\omega_t^*$ is always symmetric under permuting the $k$ registers. Given $\omega_t^*$, our actual $n$-qubit hypothesis state $\omega_t$ is then obtained by simply tracing out $k-1$ of the registers.

Given an amplified hypothesis state $\omega^*$, let $E_t^*$ be a two-outcome measurement that acts on $\omega^*$ as follows: it applies the measurement $E_t$ to each of the $k$ registers separately, and accepts if and only if the fraction of measurements that accept equals $b_t$, up to an additive error at most $\varepsilon/2$.

Here is the learning strategy. Our initial hypothesis, $\omega_0^* := \mathbb{I}/D$, is the $D$-dimensional maximally mixed state, corresponding to $\omega_0 := \mathbb{I}/2^n$. (The *maximally mixed state* corresponds to the notion of a uniformly random quantum superposition.) For each $t \geq 1$, we are given descriptions of the measurements $E_1, \ldots, E_t$, as well as real numbers $b_1, \ldots, b_t$ in $[0,1]$, such that $|b_i - \mathrm{Tr}\,(E_i\rho)| \leq \varepsilon/3$ for all $i \in [t]$. We would like to update our old hypothesis $\omega_{t-1}^*$ to a new hypothesis $\omega_t^*$, ideally such that the difference $|\mathrm{Tr}\,(E_{t+1}\omega_t) - \mathrm{Tr}\,(E_{t+1}\rho)|$ is small. We do so as follows:

- Given $b_t$, as well classical descriptions of $\omega_{t-1}^*$ and $E_t$, decide whether $\mathrm{Tr}\big(E_t^*\omega_{t-1}^*\big) \geq 1 - \frac{\varepsilon}{6}$.

- If yes, then set $\omega_t^* := \omega_{t-1}^*$ (i.e., we do not change the hypothesis).

- Otherwise, let $\omega_t^*$ be the state obtained by applying $E_t^*$ to $\omega_{t-1}^*$ and postselecting on $E_t^*$ accepting. In other words, $\omega_t^* := \mathcal{M}(\omega_{t-1}^*)$, where $\mathcal{M}$ is the operator that postselects on acceptance by $E_t^*$ (as defined above).

We now analyze this strategy. Call $t$ "good" if $\mathrm{Tr}\big(E_t^*\omega_{t-1}^*\big) \geq 1 - \frac{\varepsilon}{6}$, and "bad" otherwise. Below, we show that

(i) there are at most $\mathrm{O}\big(\frac{n}{\varepsilon^3}\log\frac{n}{\varepsilon}\big)$ bad $t$'s, and

(ii) for each good $t$, we have $|\mathrm{Tr}(E_t\omega_{t-1}) - \mathrm{Tr}(E_t\rho)| \leq \varepsilon$.

We start with claim (i). Suppose there have been $\ell$ bad $t$'s, call them $t(1), \ldots, t(\ell)$, where $\ell \leq (n/\varepsilon)^{10}$ (we justify this last assumption later, with room to spare). Then there were $\ell$ events where we postselected on $E_t^*$ accepting $\omega_{t-1}^*$. We conduct a thought experiment, in which the learning strategy maintains a quantum register initially in the maximally mixed state $\mathbb{I}/D$, and applies the postselection operator corresponding to $E_t^*$ to the quantum register whenever $t$ is bad. Let $p$ be the probability that all $\ell$ of these postselection events succeed. Then by definition,

$$p = \mathrm{Tr}\Big(E_{t(1)}^*\omega_{t(1)-1}^*\Big) \cdots \mathrm{Tr}\Big(E_{t(\ell)}^*\omega_{t(\ell)-1}^*\Big) \leq \Big(1 - \frac{\varepsilon}{6}\Big)^\ell .$$

On the other hand, suppose counterfactually that we had started with the "true" hypothesis, $\omega_0^* := \rho^* = \rho^{\otimes k}$. In that case, we would have

$$\mathrm{Tr}\Big(E_{t(i)}^*\rho^*\Big) = \mathrm{Pr}\left[E_{t(i)} \text{ accepts } \rho \text{ between } \Big(b_{t(i)} - \frac{\varepsilon}{2}\Big)k \text{ and } \Big(b_{t(i)} + \frac{\varepsilon}{2}\Big)k \text{ times}\right]$$
$$\geq 1 - 2\,e^{-2k(\varepsilon/6)^2}$$

for all $i$. Here the second line follows from the condition that $\big|\mathrm{Tr}\big(E_{t(i)}\rho\big) - b_{t(i)}\big| \leq \varepsilon/6$, together with the Hoeffding bound.

We now make the choice $k := \frac{C}{\varepsilon^2}\log\frac{n}{\varepsilon}$, for some constant $C$ large enough that

$$\mathrm{Tr}\Big(E_{t(i)}^*\rho^*\Big) \geq 1 - \frac{\varepsilon^{10}}{400n^{10}}$$

for all $i$. So by Theorem 5, all $\ell$ postselection events would succeed with probability at least

$$1 - 2\sqrt{\ell\frac{\varepsilon^{10}}{400n^{10}}} \geq 0.9 .$$

We may write the maximally mixed state, $\mathbb{I}/D$, as

$$\frac{1}{D}\rho^* + \Big(1 - \frac{1}{D}\Big)\xi ,$$

for some other mixed state $\xi$. For this reason, even when we start with initial hypothesis $\omega_0^* = \mathbb{I}/D$, all $\ell$ postselection events still succeed with probability

$$p \geq \frac{0.9}{D} \ .$$

Combining our upper and lower bounds on $p$ now yields

$$\frac{0.9}{2^{kn}} \leq \left(1 - \frac{\varepsilon}{6}\right)^{\ell}$$

or

$$\ell = \mathrm{O}\left(\frac{kn}{\varepsilon}\right) = \mathrm{O}\left(\frac{n}{\varepsilon^3} \log \frac{n}{\varepsilon}\right),$$

which incidentally justifies our earlier assumption that $\ell \leq (n/\varepsilon)^{10}$.

It remains only to prove claim (ii). Suppose that

$$\mathrm{Tr}\left(E_t^* \omega_{t-1}^*\right) \geq 1 - \frac{\varepsilon}{6} \ . \tag{8}$$

Imagine measuring $k$ quantum registers prepared in the joint state $\omega_{t-1}^*$, by applying $E_t$ to each register. Since the state $\omega_{t-1}^*$ is symmetric under permutation of the $k$ registers, we have that $\mathrm{Tr}(E_t\omega_{t-1})$, the probability that $E_t$ accepts the first register, equals the expected fraction of the $k$ registers that $E_t$ accepts. The bound in Eq. (8) means that, with probability at least $1 - \frac{\varepsilon}{6}$ over the measurement outcomes, the fraction of registers which $E_t$ accepts is within $\pm\varepsilon/2$ of $b_t$. The $k$ measurement outcomes are not necessarily independent, but the fraction of registers accepted never differs from $b_t$ by more than 1. So by the union bound, we have

$$|\mathrm{Tr}(E_t\omega_{t-1}) - b_t| \leq \frac{\varepsilon}{2} + \frac{\varepsilon}{6} = \frac{2\varepsilon}{3} \ .$$

Hence by the triangle inequality,

$$|\mathrm{Tr}(E_t\omega_{t-1}) - \mathrm{Tr}(E_t\rho)| \leq \frac{2\varepsilon}{3} + |b_t - \mathrm{Tr}(E_t\rho)| \leq \varepsilon \ ,$$

as claimed. $\qquad\square$

# E    Proof of Corollary 7

We begin with a bound for a generalization of "random access coding" (Nayak [1999], Ambainis et al. [2002]) or what is also known as the Index function problem in communication complexity. The generalization was called "serial encoding" by Nayak [1999] and arose in the context of quantum finite automata. The serial encoding problem is also called Augmented Index in the literature on streaming algorithms.

The following theorem places a bound on how few qubits serial encoding may use. In other words, it bounds the number of bits we may encode in an $n$-qubit quantum state when an arbitrary bit out of the $n$ may be recovered well via a two-outcome measurement. The bound holds even when the measurement for recovering $y_i$ may depend adaptively on the previous bits $y_1 y_2 \cdots y_{i-1}$ of $y$, *which we need not know.*

**Theorem 15** (Nayak [1999]). *Let $k$ and $n$ be positive integers. For each $k$-bit string $y := y_1 \cdots y_k$, let $\rho_y$ be an $n$-qubit mixed state such that for each $i \in \{1, 2, \ldots, k\}$, there is a two-outcome measurement $E$ that depends only on $i$ and the prefix $y_1 y_2 \cdots y_{i-1}$, and has the following properties*

    *(i)  if $y_i = 0$ then $\mathrm{Tr}(E\rho_y) \leq p$, and*

    *(ii)  if $y_i = 1$ then $\mathrm{Tr}(E\rho_y) \geq 1 - p$,*

*where $p \in [0, 1/2]$ is the error in predicting the bit $y_i$ at vertex $v$. (We say $\rho_y$ "serially encodes" $y$.) Then $n \geq (1 - \mathrm{H}(p))k$.*

In Appendix F, we present a strengthening of this bound when the bits of $y$ may be only be recovered in an adaptive order that is *a priori* unknown. The stronger bound may be of independent interest.

In the context of online learning, the measurements used in recovering bits from a serial encoding are required to predict the bits with probability bounded away from given "pivot points". Theorem 15 may be specialized to this case as in Corollary 7, which we prove below.

*Proof of Corollary 7.* This is a consequence of Theorem 15, when combined with the following observation. Given the measurement operator $E'$, parameter $\varepsilon$, and pivot point $a_v$ as in the statement of the corollary, we define a new two-outcome measurement $E$ to be associated with vertex $v$:

$$
E \quad := \quad \begin{cases} \frac{E'}{2a_v} & \text{if } a_v \geq \frac{1}{2}, \quad \text{and} \\ \frac{1}{2(1-a_v)}\left(E' + (1-2a_v)\mathbb{I}\right) & \text{if } a_v < \frac{1}{2}. \end{cases}
$$

The measurement $E$ may be interpreted as producing a fixed outcome $0$ or $1$ with some probability depending on $a_v$, and applying the given measurement $E'$ with the remaining probability, so as to translate the pivot point $a_v$ to $1/2$.

We may verify that the operator $E$ satisfies the requirements (i) and (ii) of Theorem 15 with $p := (1-\varepsilon)/2$. We therefore conclude that $n \geq (1 - \mathrm{H}((1-\varepsilon)/2)k$. Since $\mathrm{H}(1/2 - \delta) \leq 1 - 2\delta^2$, for $\delta \in [0, 1/2]$, we get $k = \mathrm{O}(n/\varepsilon^2)$. $\qquad\square$

# F   Lower bound on quantum random access codes

Here we present an alternative proof of the linear lower bound on quantum random access codes Nayak [1999], Ambainis et al. [2002]. It goes via the Matrix Multiplicative Weights algorithm, but gives us a slightly weaker dependence on decoding error. We also present an extension of the original bound to more general codes. These may be of independent interest.

**Theorem 16.** *Let $k$ and $n$ be positive integers with $k > n$. For all $k$-bit strings $y = y_1, y_2, \ldots, y_k$, let $\rho_y$ be the $n$-qubit quantum mixed state that encodes $y$. Let $p \in [0, 1/2]$ be an error tolerance parameter. Suppose that there exist measurements $E_1, E_2, \ldots, E_k$ such that for all $y \in \{0,1\}^k$ and all $i \in [k]$, we have $|\operatorname{Tr}(E_i\rho_y) - y_i| \leq p$. Then $n \geq \frac{(1/2-p)^2}{4(\log 2)}k$.*

*Proof.* Run the MMW algorithm described in Section 3.2 with the absolute loss function $\ell_t(x) := |x - y_t|$ for $t = 1, 2, \ldots, k$ iterations. In iteration $t$, provide as feedback $E_t$ and the label $y_t \in \{0, 1\}$ defined as follows:

$$
y_t = \begin{cases} 0 & \text{if } \operatorname{Tr}(E_t\omega_t) > \frac{1}{2} \\ 1 & \text{if } \operatorname{Tr}(E_t\omega_t) \leq \frac{1}{2}. \end{cases}
$$

Let $y \in \{0, 1\}^k$ be the bit string formed at the end of the process. Then it is easy to check the following two properties by the construction of the labels: for any $t \in [k]$, we have

1. $\ell_t(\omega_t) = |\operatorname{Tr}(E_t\omega_t) - y_t| \geq 1/2$, and

2. $\ell_t(\rho_y)) = |\operatorname{Tr}(E_t\rho_y) - y_t| \leq p$.

By Theorem 4, the MMW algorithm with absolute loss has a regret bound of $2\sqrt{(\log 2)kn}$. So the above bounds imply that $k/2 \leq pk + 2\sqrt{(\log 2)kn}$, which implies that $n \geq \frac{(1/2-p)^2}{4\log 2}k$. $\qquad\square$

Note that in the above proof, we may allow the measurement in the $i$th iteration, i.e., the one used to decode the $i$th bit, to depend on the previous bits $y_1, y_2, \ldots, y_{i-1}$. Thus, the lower bound also applies to serial encoding.

Next we consider encoding of bit-strings $y$ into quantum states $\rho_y$ with a more relaxed notion of decoding. The encoding is such that given the encoding for an unknown string $y$, *some* bit $i_1$ of $y$ can be decoded. Given the value $y_{i_1}$ of of this bit, a new bit $i_2$ of $y$ can be decoded, and the index $i_2$ may depend on $y_{i_1}$. More generally, given a sequence of bits $y_{i_1}y_{i_2}\ldots y_{i_j}$ that may be decoded in this manner, a new bit $i_{j+1}$ of $y$ can be decoded, for any $j \in \{0, 1, \ldots, k-1\}$. Here, the index $i_{j+1}$

and the measurement used to recover the corresponding bit of $y$ may depend on the sequence of bits $y_{i_1} y_{i_2} \ldots y_{i_j}$. We show that *even with this relaxed notion of decoding*, we cannot encode more than a linear number of bits into an $n$-qubit state.

We first formalize the above generalization of random access encoding. We view a complete binary tree of depth $d \geq 0$ as consisting of vertices $v \in \{0,1\}^{\leq d}$. The root of the tree is labeled by the empty string $\epsilon$ and each internal vertex $v$ of the tree has two children $v0, v1$. We specify an adaptive sequence of measurements through a "measurement decision tree". The tree specifies the measurement to be applied next, given a prefix of such measurements along with the corresponding outcomes.

**Definition 1.** *Let $k$ be a positive integer. A* measurement decision tree *of depth $k$ is a complete binary tree of depth $k$, each internal vertex $v$ of which is labeled by a triple $(S, i, E)$, where $S \in \{1, \ldots, k\}^l$ is a sequence of length $l := |v|$ of distinct indices, $i \in \{1, \ldots, k\}$ is an index that does not occur in $S$, and $E$ is a two-outcome measurement. The sequences associated with the children $v0, v1$ of $v$ (if defined) are both equal to $(S, i)$.*

For a $k$-bit string $y$, and sequence $S := (i_1, i_2, \ldots, i_l)$ with $0 \leq l \leq k$ and $i_j \in \{1, 2, \ldots, k\}$, let $y_S$ denote the substring $y_{i_1} y_{i_2} \cdots y_{i_l}$.

**Theorem 17.** *Let $k$ and $n$ be positive integers. For each $k$-bit string $y := y_1 \cdots y_k$, let $\rho_y$ be an $n$-qubit mixed state (we say $\rho_y$ "encodes" $y$). Suppose there exists a measurement decision tree $T$ of depth $k$ such that for each internal vertex $v$ of $T$ and all $y \in \{0,1\}^k$ with $y_S = v$, where $(S, i, E)$ is the triple associated with the vertex $v$, we have $|\mathrm{Tr}(E\rho_y) - y_i| \leq p_v$, where $p_v \in [0, 1/2]$ is the error in predicting the bit $y_i$ at vertex $v$. Then $n \geq (1 - \mathrm{H}(p))k$, where $\mathrm{H}$ is the binary entropy function, and $p := \frac{1}{k} \sum_{l=1}^{k} \frac{1}{2^l} \sum_{v \in \{0,1\}^l} p_v$ is the average error.*

*Proof.* Let $Y$ be a uniformly random $k$-bit string. We define a random permutation $\Pi$ of $\{1, \ldots, k\}$ correlated with $Y$ that is given by the sequence of measurements in the root to leaf path corresponding to $Y$. More formally, let $\Pi(1) := i$, where $i$ is the index associated with the root of the measurement decision tree $T$. For $l \in \{2, \ldots, k\}$, let $\Pi(l) := j$, where $j$ is the index associated with the vertex $Y_{\Pi(1)} Y_{\Pi(2)} \cdots Y_{\Pi(l-1)}$ of the tree $T$. Let $Q$ be a quantum register such that the joint state of $YQ$ is

$$\frac{1}{2^k} \sum_{y \in \{0,1\}^k} |y\rangle\langle y| \otimes \rho_y .$$

The quantum mutual information between $Y$ and $Q$ is bounded as $\mathrm{I}(Y : Q) \leq |Q| = n$. Imagine having performed the first $l - 1$ measurements given by the tree $T$ on state $Q$ and having obtained the correct outcomes $Y_{\Pi(1)} Y_{\Pi(2)} \cdots Y_{\Pi(l-1)}$. These outcomes determine the index $\Pi(l)$ of the next bit that may be learned. By the Chain Rule, for any $l \in \{1, \ldots, k-1\}$,

$$\mathrm{I}\big(Y_{\Pi(l)} \cdots Y_{\Pi(k)} : Q \mid Y_{\Pi(1)} Y_{\Pi(2)} \cdots Y_{\Pi(l-1)}\big)$$
$$= \mathrm{I}\big(Y_{\Pi(l)} : Q \mid Y_{\Pi(1)} Y_{\Pi(2)} \cdots Y_{\Pi(l-1)}\big) + \mathrm{I}\big(Y_{\Pi(l+1)} \cdots Y_{\Pi(k)} : Q \mid Y_{\Pi(1)} Y_{\Pi(2)} \cdots Y_{\Pi(l)}\big) .$$

Let $E$ be the operator associated with the vertex $V := Y_{\Pi(1)} Y_{\Pi(2)} \cdots Y_{\Pi(l-1)}$. By hypothesis, the measurement $E$ predicts the bit $Y_{\Pi(l)}$ with error at most $p_V$. Using the Fano Inequality, and averaging over the prefix $V$, we get

$$\mathrm{I}\big(Y_{\Pi(l)} : Q \mid Y_{\Pi(1)} Y_{\Pi(2)} \cdots Y_{\Pi(l-1)}\big) \geq \mathbb{E}_V \left(1 - \mathrm{H}(p_V)\right) .$$

Applying this repeatedly for $l \in \{1, \ldots, k-1\}$, we get

$$\mathrm{I}(Y : Q) = \mathrm{I}\big(Y_{\Pi(1)} : Q\big) + \mathrm{I}\big(Y_{\Pi(2)} : Q \mid Y_{\Pi(1)}\big) + \mathrm{I}\big(Y_{\Pi(3)} : Q \mid Y_{\Pi(1)} Y_{\Pi(2)}\big)$$
$$+ \cdots + \mathrm{I}\big(Y_{\Pi(k)} : Q \mid Y_{\Pi(1)} Y_{\Pi(2)} \cdots Y_{\Pi(k-1)}\big)$$
$$\geq \sum_{l=1}^{k} \frac{1}{2^l} \sum_{v \in \{0,1\}^l} \left(1 - \mathrm{H}(p_v)\right)$$
$$\geq (1 - \mathrm{H}(p))k ,$$

by concavity of the binary entropy function, and the definition of $p$. $\qquad\square$