[Reviews · NeurIPS 2018]

Reviewer 1



The submission studies online learning for the task of predicting the outcomes of binary measurements on copies of an n-qubit system. Specifically, the authors provide positive results based on 3 approaches: (1) FTRL algorithm; (2) Quantum Postselection; and (3) (Sequential) fat shattering dimension. (1) follows more or less the standard approach for online learning, but some mathematical ninja moves are required to deal with convexity for complex numbers. (2) gives the weakest bounds, and is presented in the least NIPS-reader-friendly way. While it is certainly a nice addition to the paper, I expect it will generate the least interest for even the quantum-curious NIPS audience. (3) seems to follow almost immediately from combining Theorems 8 and 10 from previous work. The downside is that this is non-constructive (similar to VC-type guarantees if I’m not mistaken). While the introduction does a nice job of motivating online vs offline quantum tomography, it doesn’t explain why learning quantum states is interesting to begin with. (E.g. where do all these copies of an n-qubit system come from?) PRO: It’s probably good that someone wrote this paper, and verified that FTRL indeed works in the quantum/complex setting. It will probably be a refreshing addition to the NIPS program. It’s also a good opportunity to celebrate with the community another cool application of learning theory. CON: From a learning perspective, I find the results a bit boring. Given that the problem has low dimension (3), and is convex, it is not a big surprise that FTRL works. In this sense I can imagine NIPS audience putting all the effort understanding the quantum part, and then learn very little from the learning side. QUESTIONS TO AUTHORS: 1. I didn’t quite understand what is the obstacle for achieving tight bounds for L_2 loss. Perhaps that would be more interesting than the bounds that do work. 2. The last paragraph of the submission asks whether any algorithm that produces hypothesis states that are approximately consistent with all previous data would work. I’m probably wrong, but naively that would seem to contradict the impossibility results in the first paragraph of the submission. Namely, the latter imply that after a subexp number of measurements there still exists a measurement that would separate two states consistent with everything seen so far, no? I.e. your algorithms answer according to the state that is "more likely", but answering the other way is still consistent (yet likely to be wrong). ********************************************************************* [POST REBUTTAL UPDATE]: Here is my shallow (and embarrassingly classical) understanding of your problem. 1. First, for simplicity, let's assume the space of all possible quantum state is approximated by a discrete set (this is consistent with the approximation, Lipschitz loss function, etc.). Suppose that the hidden quantum state is chosen uniformly at random from this set. 2. Now, if you could come up with a sequence of measurements that recursively agree with half of the set, then until you learned the quantum state, your algorithm cannot do better than guessing at random. 3. Therefore, you would have regret = \Theta(# of samples to learn). Since this contradicts your algorithmic result, I'm guessing that #2 is false: after a while, any measurement that you could come up with is very lopsided; i.e. one outcome agrees with almost all remaining consistent quantum states. Your algorithms predict this more likely outcome, but an algorithm that would predict the opposite outcome is bound to preform poorly. What am I missing? ********************************************************************* MINOR Page 2: “which measurements are feasible *is* dictated by Nature” Section 2 vs 3: the use of \ell vs f for the loss is a bit confusing. Section 5: It would be better to define sfat dimension before explaining how to bound it using Theorem 7.

Reviewer 2



Recently, in a breakthrough paper it has been shown by Aaronson that using ideas from computational learning theory, one can learn a n qubit quantum state using a number of measurements that grows only linearly with the number of qubits-- an exponential improvement. The assumption there was i.i.d sampling for training and testing. The current paper extends this result to adversarial/online learning settings in which i.i.d assumption is not longer safe to make. The authors generalize the RFTL and analyze its regret bound using both the standard analysis of FTRL and analysis of Matrix Multiplicative Weights method for online learning, with some tweaks required to make it work for complex matrices and approximating the Bregman divergence. The authors also prove regret bounds using the notion of sequential fat-shattering dimension. I think the paper studies and interesting question and provides novel theoretical results. The presentation of the paper was mostly clear. The claimed contributions are discussed in the light of existing results and the paper does survey related work appropriately. The paper is technically sound and the derivations seem to be correct as far as I checked (I carefully checked the regret analysis and skimmed over the sequential fat-shattering part though).

Reviewer 3



The paper gives new results in the context of online learning quantum states. Namely, there is an algorithm that can produce an estimate $\omega_t$ of n-qubit quantum state $\rho$ so that when performing any two-outcome measurement $E_t$ on the estimate, the error occurs for at most $O(n/\epsilon^2$, where $\epsilon$ is the error parameter. The paper also gives the error bound for the regret minimization model that achieves regret at most $O(\sqrt{Tn})$. What I found more interesting from the results are the implications of the results: they improve the shadow tomography result of Aaronson 2018, and the implication of any online learning for bounded sequential fat-shattering dimension will automatically give an online learning quantum state, as well as the extention of Nayak’s bound. The weakness of the paper is perhaps the algorithm is the standard regularized follow-the-leader, and the techniques of updating by sub-derivative and matrix multiplicative weights are not new. However, I think the results are quite interesting despite of the simplicity. Some minor comments: (*) I think Theorem 3 and 4 should just be Lemmas as they are not really main results. Also, I think they can be merged into one statement.

Reviewer 4



This paper provides online learning methods for estimating quantum states. The target of learning is not the quantum state \rho itself, but Tr(E\rho), a single scalar that corresponds to the outcome of a measurement E. Three different approaches are provided. The second and third approaches are largely based on the results by Aaronson in 2006 and 2007. I'm not familiar with quantum computing / quantum states. Although I found in NIPS, papers in this direction has been increasing, they are still relatively sparse. So I wish the authors can provide more background, motivating examples, and applications of this technology in the introduction, rather than talk about math from the very beginning (if space is not enough, putting some of them in appendix is still good). In the texts, there are many terms that directly appear without explanation (just to name a few): Line 72: POVM (whose abbreviation?) Line 88: quantum register Line 90: joint measurement Line 215: post-select (I know it's established in other papers, but maybe you can slightly explain why they use this verb?) Line 216: the (high-level) meaning of mapping Line 218: high level interpretation of Eq. (3)? Line 218: the definition of Tr_B in Eq. (3) Line 447: maximally mixed Again, maybe they are standard in your field, but I think slightly more explanations would greatly help most NIPS readers. Speaking of the contributions, I think the paper is about the online learnability of quantum states, whose statistical learnability is already established in Aaronson 2007. The tools used are more or less standard, and the results seem not very surprising. Given that the readability suffers a bit, I tend to put this work around borderline.

Reviewer 5



Summary: This paper considers the problem of learning mixed quantum states, in the sense of being able to predict the probability that a two-outcome measurement E would accept on the unknown quantum state rho. Prior work has considered this problem in the statistical setting, where it is assumed that there is a distribution D over two-outcome measurements, we have sample access to the distribution D, and our goal is to estimate a mixed quantum state so that with high probability for a measurement E drawn from D, the acceptance probability of the estimated state is close to the acceptance probability of the unknown state. This work considers the same problem in the online setting, where it is no longer assumed that measurements are drawn iid from some unknown distribution D. Instead, an arbitrary sequence of measurements is chosen by a (possibly adaptive) adversary, and the goal is design learning algorithms with either mistake bounds (i.e., bounds on the number of times the predicted acceptance probability is wrong my more than epsilon), or regret bounds (i.e., a bound on the cumulative loss of the predicted acceptance probabilities). The online setting is especially interesting given that the sequence of measurements being learned from is usually constrained by nature and chosen by an experimentalist (rather than being iid). The paper has two main results: The first result shows that (in the realizable setting) there exists an algorithm that receives a sequence of two-outcome measurements E_1, E_2, ... one at a time and for each outputs an estimated state omega_t such that the acceptance probability of E_t on omega_t is within epsilon of the acceptance probability of E_t on the unknown hidden state rho for all but O(n/epsilon^2) rounds. This feedback given to the algorithm comes in the form of an estimate b_t of the true acceptance probability of E_t on rho, which should be accurate to within error epsilon/3. The second result applies in the non-realizable setting, where we do not assume that the sequence of losses truly arise as the acceptance probabilities of some two-outcome measurements applied to some fixed mixed quantum state. Instead, we suppose that for each measurement E_t there is an L-Lipschitz convex loss function ell_t mapping acceptance probabilities to losses. The theorem guarantees that there exists an algorithm with regret bounded by O(L sqrt(Tn)) after T measurements, where n is the number of qubits. Section 3 shows that both of the main results can be achieved using variants of the regularized follow the leader algorithm or the matrix multiplicative weights algorithm. The regret bounds for these algorithms follow the standard proofs, however some modifications are necessary to deal with the fact that the input domain is complex. The mistake bound is a corollary of the regret bounds. Sections 4 and 5 provide alternative proofs for the two main results using different techniques. In particular, Section 4 provides an alternative mistake bound of O(n/epsilon^3 log(n/epsilon)) using repeated application of a post-selection operator (and this algorithm appears to be significantly different from the FTRL and multiplicative weights algorithms). In section 5, the authors show that a learning algorithm with regret O(L sqrt(nT) log^{3/2}(T)) exists by bounding the sequential fat shattering dimension of the prediction problem. Comments and Questions: I am relatively unfamiliar with quantum computing and the related literature, but the problem in this paper seems to be well motivated, especially the shift to the online setting from prior work using statistical assumptions on the source of measurements. The main results are clearly stated and convincing, especially given that the authors present several different techniques all arriving at similar bounds. For the broader NIPS community, I think it is interesting to see these methods applied to problems somewhat distant from more typical prediction problems. It might be helpful to include slightly more background information about the underlying problem and what the benefits of having online prediction algorithms might be. In the abstract it is mentioned that we measure multiple copies of the unknown state using the same two-outcome measurement E_1. I assume that multiple measurements are used to estimate the acceptance probabilities, since each measurement leads only to the binary accept or reject outcome. In the rest of the paper we suppose we use each measurement only once, but as feedback we get an estimate of the acceptance probability. It might be worth mentioning this again in the main body somewhere.